# Learning Human Preferences without Interaction for Cooperative AI: A Hybrid Offline-Online Approach

**Haitong Ma**
Shanghai Jiao Tong University
Shanghai Innovation Institute
Shanghai, China
`mahaitong@sjtu.edu.cn`

**Haoran Yu**
Shanghai Jiao Tong University
Shanghai, China
`harayn4@gmail.com`

**Haobo Fu**
Tencent AI Lab
Shenzhen, China
`haobofu@tencent.com`

**Shuai Li**[*]
Shanghai Jiao Tong University
Shanghai Innovation Institute
Shanghai, China
`shuaili8@sjtu.edu.cn`

## Abstract

Reinforcement learning (RL) for collaborative agents capable of cooperating with humans to accomplish tasks has long been a central goal in the RL community. While prior approaches have made progress in adapting collaborative agents to diverse human partners, they often focus solely on optimizing task performance and overlook human preferences—despite the fact that such preferences often diverge from the reward-maximization objective of the environment. Addressing this discrepancy poses significant challenges: humans typically provide only a small amount of offline, preference-related feedback and are unable to engage in online interactions, resulting in a distributional mismatch between the agent's online learning process and the offline human data. To tackle this, we formulate the problem as an $online\&offline$ reinforcement learning problem that jointly integrates online generalization and offline preference learning, entirely under an offline training regime. We propose a simple yet effective training framework built upon existing RL algorithms that alternates between offline preference learning and online generalization recovery, ensuring the stability and alignment of both learning objectives. We evaluate our approach on a benchmark built upon the Overcooked environment—a standard environment for human-agent collaboration—and demonstrate remarkable performance across diverse preference styles and cooperative scenarios.

## 1 Introduction

Humans often exhibit diverse behaviors when performing the same task, driven by individual preferences. Prior work refers to these variations as preference styles [Pugh et al., 2016, Cully and Demiris, 2017, Mao et al., 2024]. In multi-agent cooperative settings—especially multiplayer games—designing agents that can adapt to a wide range of human preferences has become a growing focus in reinforcement learning (RL) [Klien et al., 2004, Yuan et al., 2023, Dafoe et al., 2021]. To avoid the cost of human-in-the-loop training, recent research frames this challenge as an ad hoc team play [Stone et al., 2010] or a Zero-Shot Coordination (ZSC) problem [Hu et al., 2020, Mirsky et al., 2022]. However, ZSC methods typically optimize only for environment rewards, ignoring

---

[*]corresponding author

39th Conference on Neural Information Processing Systems (NeurIPS 2025).

user-specific preferences, and thus often produce behaviors misaligned with individual styles [Yu et al., 2023, Liu et al., 2025]. For instance, in *Overcooked* [Carroll et al., 2019], a human player may prefer cooking potatoes, while an agent trained solely for reward maximization might independently cook tomatoes—resulting in poor coordination despite high task performance. Moreover, ZSC assumes no prior knowledge of human behavior, limiting adaptation to preference-driven dynamics. With the rise of multiplayer gaming, the assumption that human users are fully unobservable is becoming outdated. Many games now allow players to express stylistic preferences through post-hoc feedback—most notably the "like" mechanism, where users endorse teammates after a match. These liked trajectories implicitly reflect human preferences and are widely available in real-world games such as *League of Legends*, *Honor of Kings*, and *Brawl Stars*, offering a scalable and lightweight source of preference-aligned data. This enables the training of cooperative agents that not only perform effectively but also adapt to diverse human styles.

Importantly, in many real-world games such as 5v5 matches in *League of Legends*, it is impractical to model or simulate the behavior of teammates and opponents. Therefore, our setting focuses on learning solely from the decisions of cooperative agents recorded in offline trajectories, without relying on access to or simulation of other players' strategies. We formulate this as a novel $online\&offline$ RL problem under three assumptions: (1) access to an environment that supports online interaction, (2) availability of a policy pool for training the cooperative agent, and (3) a small set of offline trajectories reflecting human preferences. The objective is to train a cooperative agent that both generalizes to unseen human partners and aligns with preferences expressed in the offline data. Unlike prior RL problems that combine online and offline learning, our setting introduces a new challenge: balancing zero-shot generalization with preference alignment. Learning proceeds in two distinct modes—online interaction for generalization and fully offline optimization for preference adaptation. The $online\&offline$ RL problem introduces several unique difficulties:

**Challenge 1: Limited preference data and no human interaction.** The number of labeled preference trajectories is small, and the agent cannot interact with the target humans online. If humans are modeled as part of the environment, this leads to unknown transition dynamics. Existing methods typically learn a reward model from limited data and optimize it online [Wirth et al., 2017, Arora and Doshi, 2021], which is infeasible here. Fully offline methods also struggle in this low-data regime—direct imitation on a few labeled trajectories often overfits and fails.

**Challenge 2: Objective mismatch between generalization and preference alignment.** Offline preference data reflect distributions that differ from those of reward-maximizing agents. Optimizing for alignment can therefore harm generalization across diverse human partners. Existing approaches, including offline preference methods [Lee et al., 2021, Liu et al., 2025, Abdelkareem et al., 2022] and ZSC methods like HSP [Yu et al., 2023] and MEP [Zhao et al., 2023], do not explicitly address this trade-off.

To address the aforementioned challenges, we draw inspiration from the pretraining and supervised fine-tuning (SFT) paradigm widely adopted in natural language processing (NLP) [Devlin et al., 2019, Radford et al., 2019, Raffel et al., 2020]. We propose a three-stage hybrid framework that integrates online generalization training with offline imitation on preference-labeled trajectories. Conceptually, the online training phase mirrors language model pretraining, while offline imitation corresponds to downstream SFT. We observe that applying online generalization recovery after offline imitation allows the pretrained agent to retain part of the learned preferences. Building upon this insight, we introduce a training strategy—Epoch-wise Alternation Recovery (EAR)—which alternates between 1-epoch lightweight behavior cloning and generalization recovery. This strategy enables cooperative agents to maintain strong generalization capabilities while effectively incorporating human preferences. To evaluate the ability for the $online\&offline$ RL problem, we further develop a dedicated benchmark based on the Overcooked environment. Extensive experiments demonstrate the effectiveness of our approach, showing consistent improvements over competitive baselines across multiple evaluation metrics.

## 2 Related work

### 2.1 Online generalization ability learning

In game AI, effective cooperation requires agents to generalize across diverse human partners, adapting to various strategies while completing tasks. This challenge mirrors that of language models,

which achieve adaptability through large-scale pretraining on diverse data [Devlin et al., 2019, Radford et al., 2019, Raffel et al., 2020]. In reinforcement learning, this generalization challenge is formalized as the ZSC problem, where agents aim to maximize returns when paired with unseen partners. The FCP framework [Strouse et al., 2021] uses policy pool to expose agents to diverse behaviors during training, while MEP [Zhao et al., 2023] extends this through entropy-maximized policy diversification. HSP [Yu et al., 2023] further enhances robustness by modeling human preferences as biased. While these ZSC methods improve generalization, they largely overlook preference learning—the ability to adapt to user-specific preferences. As a result, agents trained only for ZSC problem may struggle in personalized human-agent interactions.

## 2.2 Offline preference learning

Offline preference learning has gained increasing attention, particularly in natural language processing (NLP). Direct Preference Optimization (DPO) [Raffel et al., 2020] reframes reinforcement learning with human feedback (RLHF) [Achiam et al., 2023, Wang et al., 2024b] as a fully offline supervised objective, encouraging models to prefer higher-quality responses. Kullback-Leibler Target Optimization (KTO) [Hu and Hong, 2013] further advanced this by removing the need for explicit preference pairs, while Inverse Preference Optimization (IPO) [Azar et al., 2024] enhanced DPO's stability by refining its objective to mitigate overfitting. These methods are well-suited for large language models, which retain strong capabilities even after offline fine-tuning. In traditional RL, preference alignment has focused on two offline paradigms: offline imitation learning [Torabi et al., 2018, Kumar et al., 2020, Prudencio et al., 2023], which directly learns from offline trajectories, and preference-based approaches like offline PBRL [Wirth et al., 2017, Lee et al., 2021, Abdelkareem et al., 2022] and IRL [Ho and Ermon, 2016, Fu et al., 2017, Arora and Doshi, 2021], which aim to train a reward function from labeled data. However, both methods require large amounts of offline data, making them unsuitable for $online \& offline$ problem with limited data.

## 2.3 Online RL combined with offline data

The proposed problem requires combining online generalization with offline preference learning—a setting only partially addressed in prior RL research. Existing approaches can be grouped into three categories. The first incorporates offline data into online training via shared replay buffers to improve sample efficiency and reduce interaction costs [Song et al., 2022, Ball et al., 2023]. The second, offline-to-online RL [Nakamoto et al., 2023, Zhang et al., 2023, Wang et al., 2023], pretrains on large offline datasets and then fine-tunes online using techniques such as balanced sampling [Lee et al., 2022, Guo et al., 2023] and adaptive conservatism [Kostrikov et al., 2021, Nakamoto et al., 2023] to mitigate distributional shift [Prudencio et al., 2023, Levine et al., 2020, Andres et al., 2025]. The third, online-to-offline RL [Liu et al., 2025], begins with online training for task performance and then uses limited human feedback to fit reward models for alignment. However, these methods typically assume access to either large-scale offline data or online human interaction—assumptions that do not hold in our setting. In contrast, our multi-agent setting prohibits online interaction with users and offers only a small set of positive-only preference data, necessitating new solutions tailored to this constrained regime.

# 3 Preliminaries

## 3.1 Decentralized Markov Decision Process

A general $n$-player multi-agent game can be modeled as a *Decentralized Markov Decision Process (Dec-MDP)* [Bernstein et al., 2002, Wang et al., 2024a], defined by the tuple $\langle S, A, \rho, T, r, \gamma \rangle$. Here, $S$ is the state space, $A = A_1 \times \cdots \times A_n$ is the joint action space, and $\rho$ is the initial state distribution. The transition function $T(s' \mid s, \boldsymbol{a})$ defines the probability of reaching $s'$ from $(s, \boldsymbol{a})$, and $r(s, \boldsymbol{a})$ is the shared reward. The discount factor $\gamma \in [0, 1)$ determines the weight of future rewards. Let $\boldsymbol{\pi} = \{\pi_1, \ldots, \pi_n\}$ denote the joint policy, where each $\pi_i(a^i \mid s)$ specifies the action distribution for agent $i$. At each timestep $t$, agents act independently: $a_t^i \sim \pi_i(\cdot \mid s_t)$, yielding a joint action $\boldsymbol{a_t} = \{a_t^1, \ldots, a_t^n\}$.

### 3.1.1 Cooperative agent generalization and policy pool

In cooperative multi-agent settings, agents must coordinate their behaviors through a joint policy to maximize the cumulative return. In the ZSC problem, given a set of previously unseen human

policies $\boldsymbol{\pi^H} = \{\pi_1, \ldots, \pi_{n-1}\}$, the objective is to learn a cooperative agent $\pi^A$ that can effectively collaborate with these human partners to complete tasks and optimize the overall reward. This cooperative objective is referred to as the *generalization target*.

$$\max_{\pi^A} J_g(\pi^A, \boldsymbol{\pi^H}) = \max_{\pi^A} \mathbb{E}_{s_t, \boldsymbol{a_t} \sim \pi^A} \left[ \sum_{t=0}^{\infty} \gamma^t R(s_t, a_t^A, \boldsymbol{a_t^H}) \right], \tag{1}$$

where $s_t$ denotes the environment state at timestep $t$, and $\boldsymbol{a_t} = \{a_t^1, a_t^2, \ldots, a_t^n\}$ is the joint action taken by all $n$ agents. The subset $\boldsymbol{a_t^H} = \{a_t^1, a_t^2, \ldots, a_t^{n-1}\}$ represents the actions taken by the $n-1$ unknown human policies, and $a_t^A$ is the action taken by the cooperative agent $\pi^A$ at timestep $t$.

Since directly obtaining the true distribution of human policies $P_H$ is often infeasible, prior works [Strouse et al., 2021, Yu et al., 2023, Zhao et al., 2023] typically resort to constructing a *policy pool*—a collection of agents with diverse behaviors—to approximate this distribution, denoted as $\hat{P}_H$. The cooperative agent is then trained to collaborate with agents sampled from this policy pool, enabling it to generalize to unseen human partners drawn from the true distribution. Under this approximation, the generalization objective can be rewritten as:

$$\max_{\pi^A} J_g(\pi^A, \boldsymbol{\pi^H}) = \max_{\pi^A} \mathbb{E}_{\boldsymbol{\pi^H} \sim \hat{P}_H} \left[ J(\pi^A, \boldsymbol{\pi^H}) \right], \tag{2}$$

where the policy pool acts as a surrogate for diverse human partners. During training, we follow the setting introduced in HSP [Yu et al., 2023], where the cooperative agent interacts online with agents sampled from the policy pool. The collected trajectories are then used to update the agent using MAPPO [Yu et al., 2022] as the underlying reinforcement learning algorithm.

## 3.2 Post-match "like" assumption and preference learning objective

We assume a *post-match "like" mechanism*, where after each game episode, a human user may provide positive feedback (a "like") to one of the cooperative agents if its behavior aligns with their preferences. Accordingly, for each user $h$, we can collect a small set of positively labeled trajectories, denoted as $\left\{ \left\{ (s_t^i, a_t^i) \right\}_{t=1}^{T} \right\}_{i=1}^{N_h}$, where $T$ is the trajectory length and $N_h$ is the number of liked trajectories. Each $(s_t^i, a_t^i)$ pair represents an action at state $s_t^i$ that was endorsed by $h$.

Given only these sparse, user-specific demonstrations, we define the *preference learning objective* as the task of imitating the behaviors favored by the user. Formally, for a cooperative policy $\pi$, the goal is to maximize the likelihood of the user-approved behavior:

$$\max_{\pi^A} J_p(\pi^A) = \max_{\pi} \sum_{i=1}^{N_h} \sum_{t=1}^{T} \log \pi(a_t^i \mid s_t^i), \tag{3}$$

which corresponds to standard behavioral cloning over a limited set of positively labeled trajectories.

Crucially, although these trajectories are obtained through direct interaction with user $h$, the user is *not available* during training. Thus, the learning process must proceed entirely offline, relying solely on this small collection of feedback for preference learning.

# 4 Method

Our approach is inspired by the pretraining–SFT paradigm widely used in NLP, which bears a structural resemblance to our cooperative RL setting: generalization with a policy pool parallels large-scale self-supervised pretraining, while offline preference learning from user-approved trajectories resembles SFT on task-specific data.

Building on this insight, we propose a three-stage framework (Figure 1): (1) *Pretraining* with a policy pool to acquire generalizable cooperative behavior; (2) *Preference learning with recovery*, where offline behavior cloning on human-preferred trajectories is followed by generalization recovery to mitigate performance degradation; (3) *Epoch-wise alternation recovery*, which incrementally refines preference alignment by interleaving imitation and recovery in successive epochs.

This framework enables the agent to internalize user preferences without compromising its ability to coordinate with unseen partners. We describe each stage in detail below.

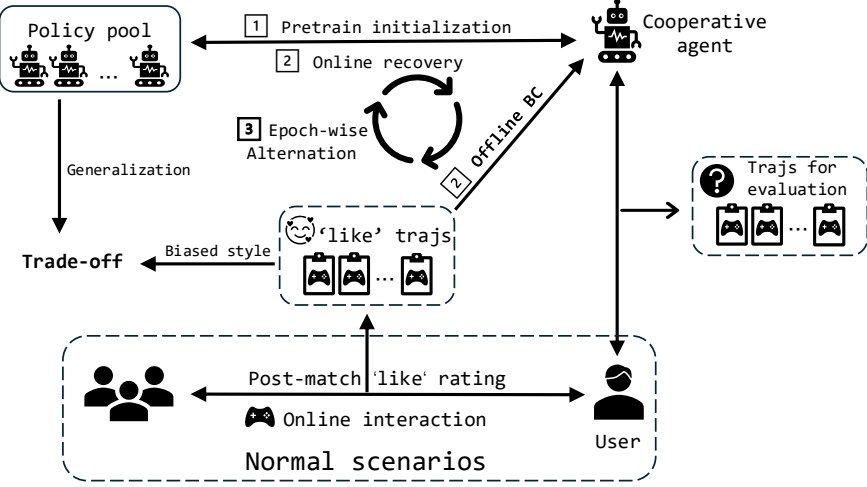

Figure 1: The *online&offline* RL problem and three-stage approach

## 4.1 Pretrain: generalization prior for cooperative agents

Our pretraining stage corresponds to the online interaction phase described in Section 3.1.1. Specifically, the cooperative agent interacts with a diverse policy pool consisting of previously deployed agents and is optimized using the MAPPO algorithm. This process aims to enable the agent to adapt to various human proxies while ensuring task completion. While conceptually similar to the pretrain-and-SFT paradigm in NLP, our setting in Game AI differs in several key aspects. Pretraining aims to enhance the model's generalization ability, but cooperative agents typically have significantly fewer parameters than language models, making their training less stable. Moreover, since the objective of preference learning is often misaligned with that of environment reward maximization, offline preference learning frequently causes a notable drop in generalization performance—referred to as "unlearning" [Kumar et al., 2020, Kostrikov et al., 2022].

Despite these challenges, pretraining is crucial. Applying preference learning to a randomly initialized agent with limited user data leads to severe overfitting. To analyze this empirically, we compare a pretrained agent and a randomly initialized agent by performing behavior cloning to convergence on a small number of offline trajectories from three distinct preference styles. After that, both agents undergo a generalization recovery phase via online interaction with the policy pool. As shown in Figure 2, the blue curve represents the agent's average environment return when cooperating with partners from the policy pool. The results indicate that achieving comparable performance without pretraining requires substantially more training. Furthermore, when evaluated under specific preference styles, agents trained from scratch often fail to recover the generalization performance of their pretrained counterparts, even after extensive optimization. This suggests that, for smaller models, pretraining not only enhances data efficiency but also prevents irreversible loss of generalization. Overfitting to user preferences from scratch may impair the agent's ability to exhibit broadly cooperative behavior.

## 4.2 Preference learning with recovery: empirical analysis of the conflict between offline behavioral cloning and online generalization

To simultaneously pursue preference learning convergence and preserve generalization ability (i.e., maintaining performance comparable to the agent initialized from pretraining), as shown in Algorithm 1 the second stage of our algorithm first performs offline behavior cloning on human-preferred trajectories, followed by a generalization recovery phase to mitigate potential performance degradation. Although the optimization objectives of these two procedures are not entirely consistent, our experiments in this section demonstrate that such a two-step design helps achieve a better balance between preference learning and generalization preservation.

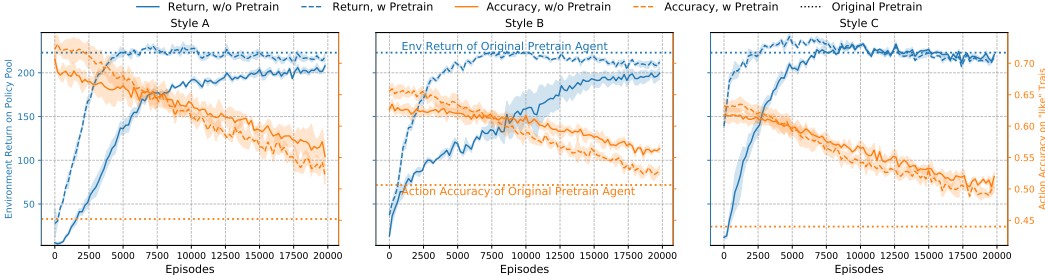

Figure 2: Recovery dynamics after behavior cloning. Dashed lines indicate pretrained agents, while solid lines indicate agents trained from scratch (no pretraining). Blue curves represent the average environment return over the policy pool, and orange curves represent the action accuracy of the same pretrained agent evaluated on three distinct preference datasets ("like" datasets). Horizontal dotted lines denote the initial performance of the pretrained agent.

To better understand the optimization dynamics between preference learning and generalization, we further analyze their interaction using the same experimental setup described earlier. In addition to environment return, we also track the agent's action agreement accuracy on the "liked" trajectories, which indicates how well the agent maintains alignment with user preferences throughout training.

As illustrated in Figure 2, we visualize the trade-off using both metrics. We observe that when a well-generalized cooperative agent is trained to convergence on the preference dataset, its environment return can drop by one-third or more when evaluated with the policy pool. In contrast, recovering generalization is considerably slower, as the agent must relearn how to cooperate with diverse partners across a range of game situations.

Empirically, fully recovering generalization often requires tens of thousands episodes of online training. Moreover, this recovery process can partially degrade the agent's preference learning performance. Interestingly, after the recovery phase, the agent does not entirely forget its prior alignment. Its accuracy on the "liked" trajectories remains higher than that of the initial pretrained agent, indicating that some preference information is retained. We observe a similar phenomenon under lightweight preference learning: even with only one epoch of behavioral cloning, the agent retains partial preference learning after the recovery phase. Detailed results are presented in appendix A. This observation motivates the third stage of our approach—an alternating training strategy that consistently interleaves preference learning with generalization recovery to achieve a stable and effective integration of both objectives.

### 4.3 Epoch-wise alternation recovery

Based on the above observations, we propose epoch-wise alternation recovery, where each round consists of one epoch of BC followed by generalization recovery. This strategy allows for further improving preference alignment while preserving generalization.

Building on the three stages of our approach, we outline a practical training algorithm (see Algorithm 1). Starting from a pretrained agent that has converged on the policy pool (Lines 3), we first record its average environment return as the generalization baseline. In the second stage, we perform sufficient behavior cloning on the preference trajectories until convergence and record the corresponding accuracy on the offline dataset (Line 4-5). We then recover the agent's generalization ability to match that of the original pretrained agent. In the third stage, we apply epoch-wise alternation recovery—repeating one epoch of BC followed by recovery—until the preference accuracy reaches the level obtained after full convergence in the second stage (Lines 6–10).

This epoch-wise alternation cycle is repeated until either the preference accuracy recovers to the level achieved in the second stage or a predefined maximum number of iterations is reached. By adopting this shallow and incremental training scheme, we enable gradual improvement in both generalization and preference alignment, while maintaining overall training stability.

---

**Algorithm 1** Epoch-wise alternation recovery

---

1: **Input:** Policy pool $\{\hat{\pi}_j^H\}_{j=1}^M$, offline preference data $\mathcal{D}$, environment $\mathcal{E}$, initial agent $\pi$;
2: **Output:** Well-trained cooperative agent $\pi$;
3: Train $\pi$ with $\{\hat{\pi}_j^H\}$; record average return $r_{gen}$; # Stage 1
4: BC train $\pi$ on $\mathcal{D}$ until convergence; record accuracy $\eta$; # Stage 2
5: Recover $\pi$ with policy pool on $\mathcal{E}$ until avg. return $\geq r_{gen}$ or budget exhausted;
6: **repeat** # Stage 3: epoch-wise alternation recovery
7:     Train $\pi$ on $\mathcal{D}$ for 1 epoch BC;
8:     Recover $\pi$ with policy pool on $\mathcal{E}$ until avg. return $\geq r_{gen}$ or budget exhausted;
9:     Evaluate accuracy $\eta_{curr}$ on $\mathcal{D}$;
10: **until** $\eta_{curr} \geq \eta$ or budget exhausted;
11: **return** Final policy $\pi$

---

## 5 Experiments

In this section, we aim to answer the following research questions through the experimental evaluations: (1) Can the 3-stage approach effectively align the cooperative agent's behavior with that of a target human proxy, producing the desired interaction style? (2) Beyond the specific human proxy used for preference learning, does the aligned agent retain its ability to collaborate effectively with other unseen human proxies? (3) Is the *epoch-wise alternation recovery* necessary, and how does it influence the agent? Can the this stage be completed within a limited training budget?

### 5.1 Environment and human proxy with different preferences

Following prior works [Strouse et al., 2021, Yu et al., 2023, Zhao et al., 2023], we adopt the Overcooked environment [Carroll et al., 2019] to simulate two-player cooperation. In this setting, a cooperative agent and a human proxy must coordinate to complete cooking tasks within a time limit. Agents perform actions such as moving, picking up, and delivering ingredients to prepare and serve soups. Success depends on coordination rather than individual skill—agents that perform well in self-play may still fail to collaborate effectively with a human proxy. More details about the environment will be shown in appendix B.

To simulate user preferences, we follow the framework used in HSP and ZSC-Eval, modeling preferences as latent *preference reward functions* that deviate from the environment-defined reward. As human preferences are typically *event-centric*, we induce stylistic diversity by assigning biased weights to specific gameplay events. Formally, we define the reward class as:

$$\mathcal{R} = \{R_w : R_w(s, a_1, a_2) = \phi(s, a_1, a_2)^\top w, \ |w|\infty \leq C\text{max}\}, \tag{4}$$

where $\phi(s, a_1, a_2)$ denotes features of state-action pairs and $w$ encodes preference bias. A sampled reward $R_w$ induces a *human proxy policy* by reaching a Nash equilibrium via self-play with a standard cooperative agent, yielding a partner policy with consistent, biased preferences.

### 5.2 Benchmark, baselines, evaluation protocols

**Benchmark.** Following the assumptions in HSP [Yu et al., 2023], we construct a set of $N$ biased reward functions $\{R_w^i\}_{i=1}^N$. For each $R_w^i$, we train a policy pair $(\pi_h^i, \pi_a^i)$ via self-play to reach a Nash equilibrium. Each pair is used to generate two trajectories, yielding a $2N$-sized offline dataset. To define human preferences, we randomly sample three biased reward functions from $\{R_w^i\}$ and use each to evaluate all trajectories. For each sampled function, we retain the trajectories whose biased return reaches at least 80% of the maximum return under that preference, and treat them as *liked* trajectories. The corresponding human-side policies $\{\pi_h^i\}$ in these trajectories serve as test-time collaborators for evaluating the cooperative agent. To better simulate the diversity of human preferences, we employ multiple stylistically similar human proxies to introduce behavioral variability. For both pretraining and recovery, we adopt the same policy pool as used in the FCP [Strouse et al., 2021], which comprises suboptimal checkpoints from self-play under a neutral reward. This ensures that preference adaptation arises purely from the offline trajectories. We conduct experiments on two Overcooked layouts (*Coordination Ring*, *Many Orders*), with detailed layout configurations provided

in Appendix B. For each setting, results are averaged over three runs with different random seeds to ensure robustness. Hyperparameter details are provided in Appendix C.

**Baselines** We denote the benchmark-provided golden trajectories as **Offline Trajs**. We compare the proposed method against the following baselines commonly used in preference-aware multi-agent reinforcement learning: (1) **Behavior Cloning (BC)** [Torabi et al., 2018], which directly trains an agent via behavior cloning on the offline preference trajectories without any pretraining; (2) **ZSC (Pretrain)** [Strouse et al., 2021], where a generalizable cooperative agent is trained using FCP, a representative ZSC method. To control for the effect of implicit preference alignment during pretraining, we intentionally select a sub-optimal checkpoint; (3) **BC on Pretrained Agent (Pretrain + BC)**, which initializes the agent with the FCP-pretrained agent and further fine-tunes it using behavior cloning on the offline preference data; (4) **BC and Recovery on Pretrained Agent (Pretrain + BC + Recover)**, which first applies behavior cloning on the FCP-pretrained agent and subsequently performs online generalization recovery using interaction with a diverse policy pool.

**Evaluation protocols.** The evaluation focuses on the cooperative agent's performance when paired with human proxies in rollouts. We propose two metrics aligned with our dual objectives of task competence and preference learning:

- **Environment Return.** This metric evaluates the agent's task completion capability in the Overcooked environment, quantified by the total number of successfully delivered soups. Specifically, $\text{EnvReturn} = 20 \times N_{\text{delivered}}$, where $N_{\text{delivered}}$ denotes the number of completed deliveries.

- **Preference Score.** This metric measures the agent's alignment with human preferences, modeled as event-centric rewards. Each preference model assigns biased weights to key gameplay events such as picking up ingredients, cooking, or delivering dishes. The preference score is computed as:

$$\text{PreferenceScore} = \sum_i \text{freq}(e_i) \cdot \text{score}(e_i),$$

where $e_i$ denotes the $i$-th event type, $\text{freq}(e_i)$ its observed frequency, and $\text{score}(e_i)$ the corresponding reward weight determined by the preference function $R_w$. Detailed event definitions and scoring schemes are provided in Appendix B.

### 5.3 Preference learning performance (RQ1)

The main experimental results are summarized in Table 1. The two metrics respectively evaluate the agent's generalization capability and its alignment with user preferences given limited supervision. The results highlight the importance of pretraining: agents trained purely via BC on a small offline dataset often fail to complete tasks effectively. Likewise, ZSC-based methods trained without access to preference-specific data tend to exhibit generic behavior styles and struggle to coordinate efficiently with human agents whose actions deviate from those seen in the policy pool. Simply fine-tuning a pretrained agent via BC results in unstable performance, with some preference styles leading to degraded task execution. This further emphasizes the necessity of incorporating a recovery phase to restore generalization. In addition, we conducted supplementary experiments with diversity-oriented ZSC approaches (MEP [Zhao et al., 2023] and TrajDi [Lupu et al., 2021]), which primarily enhance policy pool diversity without leveraging offline preference data. Their relatively weaker performance on our benchmark further highlights the necessity of incorporating offline preference learning. Detailed results are provided in Appendix F.

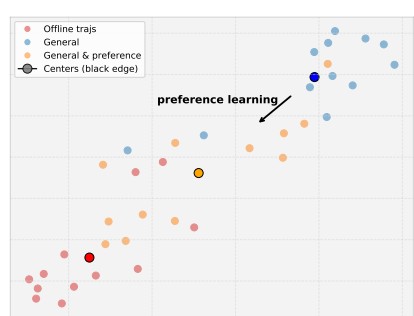

Figure 3: UMAP visualization for event-count vectors from Style A. The centers of three clusters are denoted as black edges.

At the behavioral level, we further analyze the agent's stylistic alignment by computing event-count vectors from trajectories generated by the offline dataset, the pretrained cooperative agent, and our method. We then apply UMAP for dimensionality reduction and visualization. As shown in Figure 3, trajectories generated by our method (orange points) are closer to those in the offline dataset (red

Table 1: Performance across different preference styles. Each method is evaluated using Preference Score (↑) and Environment Return (↑). Results are reported on the layout *Coordination Ring*; results on *Many Orders* are provided in Appendix D.

| Method | Style A | | Style B | | Style C | |
|---|---|---|---|---|---|---|
| | Pref. Score | Env. Return | Pref. Score | Env. Return | Pref. Score | Env. Return |
| Offline Trajs | 670.0 | 225.0 | 235 | 121 | 552 | 224 |
| Pretrain | $111.3 \pm 6.6$ | $47.6 \pm 1.2$ | $129.3 \pm 3.8$ | $60.9 \pm 2.6$ | $188.7 \pm 6.2$ | $84.0 \pm 3.4$ |
| Pretrain + BC | $203.5 \pm 39.7$ | $53.2 \pm 12.8$ | $178.0 \pm 33.7$ | $82.5 \pm 20.8$ | $120.7 \pm 50.0$ | $34.6 \pm 15.8$ |
| BC Only | $95.4 \pm 34.9$ | $13.7 \pm 7.1$ | $97.7 \pm 20.2$ | $31.9 \pm 12.5$ | $44.2 \pm 42.3$ | $4.6 \pm 6.5$ |
| Pretrain + BC + Recover | $224.3 \pm 66.0$ | $82.0 \pm 24.2$ | $187.7 \pm 18.5$ | $96.3 \pm 10.9$ | $252.0 \pm 9.41$ | $109.7 \pm 3.8$ |
| **EAR** | $\mathbf{274.7} \pm 74.0$ | $\mathbf{88.6} \pm 31.0$ | $\mathbf{200.3} \pm 17.0$ | $\mathbf{103.6} \pm 9.4$ | $\mathbf{259.0} \pm 9.9$ | $\mathbf{110.3} \pm 3.3$ |

points), indicating better alignment with user preferences. Moreover, by comparing the centroids of each distribution, we observe a clear shift from the pretrained agent's behavior (blue points) toward that of the offline datasets. This directional shift provides additional evidence that our method effectively integrates stylistic preferences into the cooperative agent's behavior. More detailed case analyses are provided in the Appendix E.

## 5.4 Generalization ability (RQ2)

Beyond preference learning, we evaluate the agent's task performance by measuring its environment return when cooperating with various style-specific proxy agents. As shown in Table 1, compared to the agent trained with standard pretraining alone, our method achieves strong performance when cooperating with the proxy agents used to generate the offline trajectories. In certain cases, such as style B, applying behavior cloning directly on the pretrained agent also yields competitive performance. To verify the necessity of the recovery phase and to more comprehensively evaluate the agent's generalization ability, we further test the cooperative agents against benchmark proxy agents that were not involved in the offline preference

Table 2: Average environment returns (↑) across all human proxy agents $\pi_h^i$ (the $N$ biased human proxies defined in the benchmark) under different methods and preference styles. As the pretrained agent is style-agnostic, its performance is reported only once.

| Preference | BC | Pretrain + BC | EAR |
|---|---|---|---|
| Style A | $5.1 \pm 2.3$ | $14.6 \pm 3.6$ | $\mathbf{86.2} \pm 12.6$ |
| Style B | $12.1 \pm 5.0$ | $31.3 \pm 9.7$ | $\mathbf{77.1} \pm 2.5$ |
| Style C | $3.0 \pm 4.2$ | $28.6 \pm 19.8$ | $\mathbf{73.9} \pm 8.5$ |
| Pretrain | | $72.7 \pm 0.1$ | |

data. The results in Table 2 show that our method maintains generalization ability comparable to the original ZSC-trained agents, despite being fine-tuned for specific human preferences. In contrast, the pretraining followed by supervised fine-tuning variants suffer a sharp drop in generalization performance. This highlights that our method effectively balances preference alignment and generalization, and further underscores the necessity of the recovery phase.

## 5.5 Impact and feasibility of epoch-wise alternation recovery (RQ3)

In this section, we validate the necessity of the *epoch-wise alternation recovery*. As shown in Table 1, compared to the conventional two-stage approach—pretraining followed by behavioral cloning and generalization recovery—our method achieves higher preference scores. To further examine the learning dynamics under different training paradigms, we compare the action accuracy on the offline "liked" trajectories after recovery. The comparative results are presented in Table 3, as consistent with Section 4.2, we observe that

Table 3: Action accuracy (↑) of behavior cloning under different training methods and preference styles.

| Method | Style A | Style B | Style C |
|---|---|---|---|
| Pretrain + BC | $0.73 \pm 0.02$ | $0.64 \pm 0.01$ | $0.64 \pm 0.01$ |
| Recover | $0.66 \pm 0.03$ | $0.59 \pm 0.05$ | $0.63 \pm 0.01$ |
| EAR | $0.73 \pm 0.02$ | $0.64 \pm 0.01$ | $0.64 \pm 0.01$ |

applying generalization recovery after fully converged BC can significantly degrade action accuracy on the "liked" trajectories—by up to 10% under certain preference styles (e.g., A and B). In contrast, *epoch-wise alternation recovery*, which interleaves one epoch of BC with a recovery phase, enables the agent to incrementally restore action accuracy while maintaining generalization capability. Notably, for preference styles that suffer greater accuracy drop, this strategy leads to more substantial

improvements in preference scores, highlighting the necessity of *epoch-wise alternation recovery* in such scenarios. These results further demonstrate that even parameter-efficient agents can effectively acquire and retain human preferences when trained through a stable and iterative alternation process.

Table 4: Number of rollouts performed when the EAR algorithm converges under different preference styles in the *Coordination Ring* layout (maximum *rollout batch* = 250).

| Max=250 | Style A | Style B | Style C |
|---|---|---|---|
| EAR | $204.3 \pm 30.0$ | $188.3 \pm 26.4$ | $205.7 \pm 24.6$ |

Regarding algorithmic feasibility, our design ensures that the EAR algorithm exhibits stable convergence in practice. During the *epoch-wise alternation recovery* phase, training proceeds iteratively, where each *rollout batch* contains 200 trajectories, and each trajectory spans 400 environment steps. This configuration yields approximately $200 \times 400 = 8 \times 10^4$ steps per batch, with a total environment interaction budget of $2 \times 10^6$ steps throughout the entire training process. At the end of each rollout batch, we evaluate the cooperative agent's environment return over the policy pool; if the criterion is met, preference accuracy is assessed (see Algorithm 1). We record the number of rollout batches required for convergence under three distinct preference styles in the *Coordination Ring* layout, as summarized in Table 4. The results confirm that the EAR algorithm converges reliably within a reasonable training budget, demonstrating its practical feasibility.

## 6 Conclusion and future work

In this paper, motivated by practical demands in the gaming industry, we propose a novel reinforcement learning problem that jointly considers generalization ability and preference alignment. Drawing inspiration from the pretraining followed by supervised fine-tuning paradigm in NLP, we design a 3-stage training framework that integrates online generalization learning with offline preference modeling. Furthermore, we introduce an alternating optimization strategy to stabilize the training process. Our method is empirically validated through extensive experiments across diverse scenarios, demonstrating its effectiveness. While current work focuses on the Overcooked environment—which is representative yet relatively simple—we plan to extend our approach to more complex environments in the future. Additionally, we aim to explore applications beyond the gaming domain, broadening the impact and applicability of the $online\&offline$ RL problem.

## 7 Acknowledgments

The corresponding author Shuai Li is sponsored by CCF-Tencent Open Research Fund.

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

## A    Lightweight behavior cloning and online recovery analysis

We further analyze the interplay between lightweight behavior cloning and online recovery. On the random1 layout, we start with a pretrained agent and conduct 1-epoch behavior cloning on the "liked" trajectories for each of the three preference styles. This is followed by online recovery using the policy pool. The corresponding training curves are shown in Figure 4.

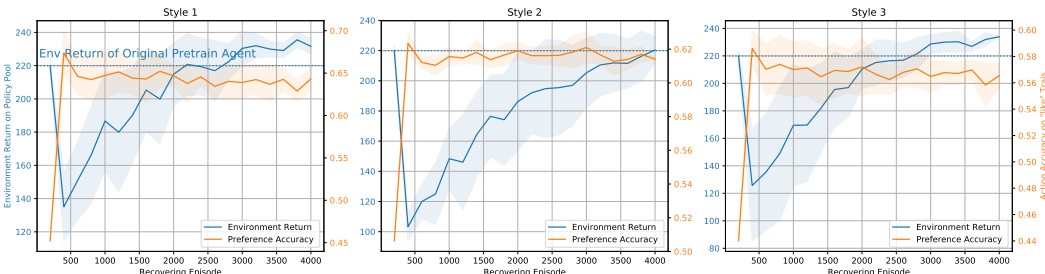

Figure 4: Training curve of online recovery after one epoch of preference behavior cloning. The blue horizontal line indicates the environment return of the pretrained agent on the policy pool. The yellow curve shows the agent's action accuracy on the offline "liked" trajectories during recovery.

Our results reveal that even a minimal application of offline preference learning—namely, a single round of behavior cloning on a small dataset—can lead to a substantial degradation in generalization performance, with environment returns dropping by more than one-third. This highlights a clear inefficiency mismatch between behavior cloning and recovery, where the latter requires thousands of online interaction episodes to restore the agent's original performance level.

Interestingly, we observe that after recovery, the agent not only regains its generalization ability but also retains partial alignment with the offline "liked" trajectories. This indicates that preference information is not entirely forgotten during recovery, providing empirical support for the effectiveness of our *Epoch-wise Alternation Recovery* in reinforcing preference learning while maintaining generalization.

## B    Overcooked benchmark details

The Overcooked environment[Carroll et al., 2019] is a widely adopted benchmark for evaluating cooperative behaviors in reinforcement learning. Inspired by the multiplayer video game Overcooked!, it simulates a kitchen where two players, each controlling a chef, must coordinate within a confined space and under time constraints to prepare and deliver soups. Upon successful delivery, both players receive a shared reward. The primary objective is to maximize the cumulative team reward through effective collaboration.

To prepare a soup, agents typically follow a multi-step process: (1) collect and place the correct ingredients into a pot according to the recipe; (2) wait for the soup to cook for a fixed duration; and (3) serve the cooked soup on a dish and deliver it to the designated location. Each agent operates in a discrete action space: up, down, left, right, interact, no-op. Beyond learning the mechanics of soup preparation, agents must also adapt to the behavioral tendencies and preferences of their partners. High team performance requires precise coordination and mutual understanding between agents.

### B.1    Layouts

Our work builds upon the Overcooked version used in the HSP benchmark[Yu et al., 2023], primarily focusing on two layouts: Coordination Ring and Many Orders.

As shown in the left part of Figure 5 , Coordination Ring features a ring-shaped kitchen layout. The tightly arranged space facilitates faster soup preparation, with ingredients, serving station, and plates all located in the bottom-left corner, and two pots positioned in the top-right. The key coordination challenge in this layout lies in the agents' movement directions to avoid collisions. The recipe is

Figure 5: Two layouts in our experiments.

relatively simple—only onion soup is required—and each soup takes 20 time steps to cook, yielding a reward of 20.

Many Orders, in contrast, adopts a square-shaped kitchen and introduces greater recipe complexity. It includes three types of orders: onion, tomato, and a mixed recipe requiring 1 onion and 2 tomatoes. Each recipe provides a different reward. In this layout, the central coordination challenge is recipe selection. If the human partner shows a preference for a particular recipe, the cooperative agent is expected to align with that preference to achieve satisfactory collaboration.

## B.2 Events

Following the settings established in HSP, we adopt the same event taxonomy to define key sub-events within each layout. For both layouts (Coordination Ring and Many Orders), the primary sub-events and their corresponding event scores are listed as following Table 12,6 :

Table 5: Event catagories and responding reward weights of Layout Coordination Ring

| Event ($e_i$) | Event reward weight (score($e_i$)) |
|---|---|
| Picking up an onion from the onion dispenser | -10, 0, 10 |
| Picking up a dish from the dish dispenser | 0, 10 |
| Picking up a ready soup from the pot with a dish | -10, 0, 10 |
| Placing an onion into the pot | -10, 0, 10 |
| Delivering a soup to the serving area | -10, 0 |

Table 6: Event catagories and responding reward weights of Layout Many Orders

| Event ($e_i$) | Event reward weight (score($e_i$)) |
|---|---|
| Picking up an onion from the onion dispenser | -10, 0, 10 |
| Picking up a dish from the dish dispenser | 0, 10 |
| Picking up a tomato from tomato dispenser | 0, 10, 20 |
| Picking up a soup | -5, 0, 5 |
| Viable placement | -10, 0, 10 |
| Optimal placement | -10, 0 |
| Catastrophic placement | 0, 10 |
| Placing an onion into the pot | -3, 0, 3 |
| Placing a tomato into the pot | -3, 0, 3 |
| Delivering a soup to the serving area | -10, 0 |

### B.3 Offline "like" trajectories

For the offline datasets, we collect trajectories across two layouts and three preference styles, with the number of trajectories for each combination summarized in Table 7. For each layout-style pair, we split the trajectories into training and test sets using a 4:1 ratio. In our reported results, action accuracy refers to performance evaluated on the test set.

Table 7: Number of trajectories in the offline datasets across two layouts and three preference styles. Each set is split into training and test sets with a 4:1 ratio.

| Layout | Style A | Style B | Style C | total |
|---|---|---|---|---|
| Coordination Ring | 12 | 24 | 20 | 72 |
| Many Orders | 38 | 14 | 12 | 72 |

## C Hyper parameters

n this section, we present the hyperparameters used and compute resources in our experiments.

### C.1 Agent architecture

Following the experimental settings in HSP, we adopt a CNN-MLP based architecture for all experiments. The convolutional module consists of three layers with output channels {32, 64, 32}. Each layer uses a kernel size of 3, padding of 1, and stride of 1. A max pooling layer is applied after the convolutional block, followed by a two-layer MLP that outputs the final action logits.

### C.2 FCP policy pool

To construct the policy pool, we perform self-play training with the objective of maximizing environment rewards. A total of 6 self-play runs are conducted, and for each run, we select three checkpoints—initial, middle, and final—resulting in a policy pool containing 18 diverse agents. The hyperparameters used for the self-play procedure are listed in Table 8 :

Table 8: Hyperparameters of self-play process

| Hyperparameters | Values |
|---|---|
| Entropy coef | 0.01 |
| Gradient clip norm | 10.0 |
| GAE lambda | 0.95 |
| Gamma | 0.99 |
| Value loss | huber loss |
| Huber delta | 10.0 |
| Mini batch size | batch size / mini-batch |
| Optimizer | Adam |
| Optimizer epsilon | 1e-5 |
| Weight decay | 0 |
| Network initialization | Orthogonal |
| Use reward normalization | True |
| Use feature normalization | True |
| Learning rate | 5e-4 |
| Parallel environment threads | 100 |
| Ppo epoch | 15 |
| Environment steps | 10M |
| Episode length | 400 |
| Reward shaping horizon | 100M |

## C.3 Pretraining, recovery, behavior cloning

Since pretraining and online generalization recovery share the same underlying procedure, we adopt identical hyperparameters for both stages. Additionally, the hyperparameters used for behavior cloning during preference learning are also summarized in Table 9.

Table 9: Hyperparameters of pretraining, online recovery, behavior cloning.

| Hyperparameters | Values |
|---|---|
| BC batch size | 32 |
| BC epoch at stage 2 | 16 |
| Hyperparameters | Values |
| N training threads | 1 |
| N rollout threads | 200 |
| Entropy coef | 0.01 |
| Gradient clip norm | 10.0 |
| GAE lambda | 0.95 |
| Gamma | 0.99 |
| Value loss | huber loss |
| Huber delta | 10.0 |
| Mini batch size | batch size / mini-batch |
| Optimizer | Adam |
| Optimizer epsilon | 1e-5 |
| Weight decay | 0 |
| Network initialization | Orthogonal |
| Use reward normalization | True |
| Use feature normalization | True |
| Learning rate | 5e-4 |
| Parallel environment threads | 100 |
| Ppo epoch | 15 |
| Environment steps | 20M |
| Episode length | 400 |
| Reward shaping horizon | 100M |

# D  Results on the Many Orders Layout

Table 10: Performance across different preference styles. Each method is evaluated using Preference Score (↑) and Environment Return (↑). Results are reported on the layout *Many Orders*

| Method | Style A | | Style B | | Style C | |
|---|---|---|---|---|---|---|
| | Pref. Score | Env. Return | Pref. Score | Env. Return | Pref. Score | Env. Return |
| Offline Trajs | 244.7 | 203.4 | 669.3 | 336.4 | 1002.5 | 378.3 |
| Pretrain | $194.3 \pm 18.0$ | $166.7 \pm 16.0$ | $124.2 \pm 17.4$ | $125.0 \pm 29.7$ | $227.3 \pm 92.1$ | $120.2 \pm 63.0$ |
| Pretrain + BC | $145.0 \pm 16.8$ | $116.4 \pm 13.8$ | $\mathbf{334.0} \pm 4.5$ | $\mathbf{340.0} \pm 2.2$ | $616.3 \pm 27.1$ | $337.3 \pm 15.2$ |
| BC Only | $86.3 \pm 51.9$ | $62.8 \pm 41.5$ | $314.0 \pm 9.9$ | $316.3 \pm 11.0$ | $\mathbf{661.3} \pm 13.5$ | $\mathbf{363.0} \pm 8.0$ |
| Pretrain + BC + Recover | $\mathbf{213.7} \pm 5.7$ | $\mathbf{178.7} \pm 4.6$ | $228.3 \pm 44.1$ | $230.3 \pm 59.9$ | $390.7 \pm 102.9$ | $217.0 \pm 54.3$ |
| **EAR** | $212.7 \pm 4.5$ | $178.3 \pm 4.2$ | $239.0 \pm 53.8$ | $239.3 \pm 68.8$ | $395.3 \pm 117.1$ | $222.3 \pm 65.8$ |

Table 11: Average environment returns (↑) across all benchmark human proxy agents $\pi_h^i$ under different methods and preference styles in layout *Many Orders*. Since the pretrained agent is style-agnostic, its performance is reported only once.

| Preference | BC | Pretrain + BC | EAR |
|---|---|---|---|
| Style A | $33.6 \pm 21.6$ | $14.6 \pm 3.6$ | $\mathbf{146.3 \pm 7.9}$ |
| Style B | $68.2 \pm 4.5$ | $86.5 \pm 3.5$ | $\mathbf{172.3 \pm 26.0}$ |
| Style C | $65.0 \pm 2.9$ | $67.9 \pm 2.6$ | $\mathbf{195.3 \pm 21.1}$ |
| Pretrain | | $142.0 \pm 22.9$ | |

In this section, we present the experimental results on the Many Orders layout. All results are averaged over three runs with different random seeds. As shown in Table 10, our proposed method, EAR, achieves consistent improvements across both metrics compared to standard BC and Recovery on Pretrained Agent. However, we observe that for certain styles (e.g., Style B and C), simple behavior cloning can already achieve excellent performance. This is largely due to the fact that partners in this benchmark exhibit strong adaptability—a favorable condition that is not commonly observed and has been refuted by results in Style A and the Coordination Ring layout. Overall, our method demonstrates robust and competitive performance across diverse settings, highlighting its effectiveness under more realistic and challenging scenarios.

Furthermore, when cooperating with proxy agents that are distinct from the "like" trajectories in the benchmark, we investigate the generalization capability of collaborative agents. Table 11 reveals that while standard BC leads to overfitting on offline trajectories—resulting in degraded performance on out-of-distribution scenarios—our method maintains its effectiveness. This finding underscores the importance of the recovery in enabling generalization.

# E  Case analysis

In this section, we provide a behavioral analysis of the agent trained with EAR, focusing on its coordination patterns under *Layout Coordination Ring* with *Style A*, which corresponds to Figure 3. We compare three types of trajectories: (1) the human-preferred offline trajectory, (2) the generalizable pretrained agent, and (3) the EAR-trained agent that balances generalization and preference alignment. The preference vector for Style A is summarized in Table 12, emphasizing actions such as "picking up a dish from the dish dispenser" and "picking up an onion from the onion dispenser." In essence, the human proxy tends to favor trajectories that include these two operations while completing the cooperative cooking task.

For each method, we select a representative trajectory and visualize frames sampled every 40 timesteps from the first 200 steps. As shown in Figure 6, the offline human proxy (top row) primarily stays near the dish and onion dispensers, focusing on delivering items to the teammate—demonstrating a clear division of labor consistent with the stated preferences. In contrast, the pretrained agent (middle row), trained solely for reward maximization, fails to account for such stylistic tendencies. When the pot is nearly ready, it prematurely moves to pick up a dish for serving, thereby blocking its partner and disrupting coordination. The EAR-trained agent (bottom row) achieves a more balanced behavior: it respects the human proxy's preferred roles while maintaining adaptive flexibility, taking initiative to fetch dishes when appropriate. This illustrates how EAR effectively reconciles the trade-off between preference alignment and generalization.

Table 12: Event catagories and responding reward weights of Layout Coordination Ring

| Event ($e_i$) | Event reward weight (score($e_i$)) |
|---|---|
| Picking up an onion from the onion dispenser | 10 |
| Picking up a dish from the dish dispenser | 10 |
| Picking up a ready soup from the pot with a dish | -10 |
| Placing an onion into the pot | 0 |
| Delivering a soup to the serving area | -10 |

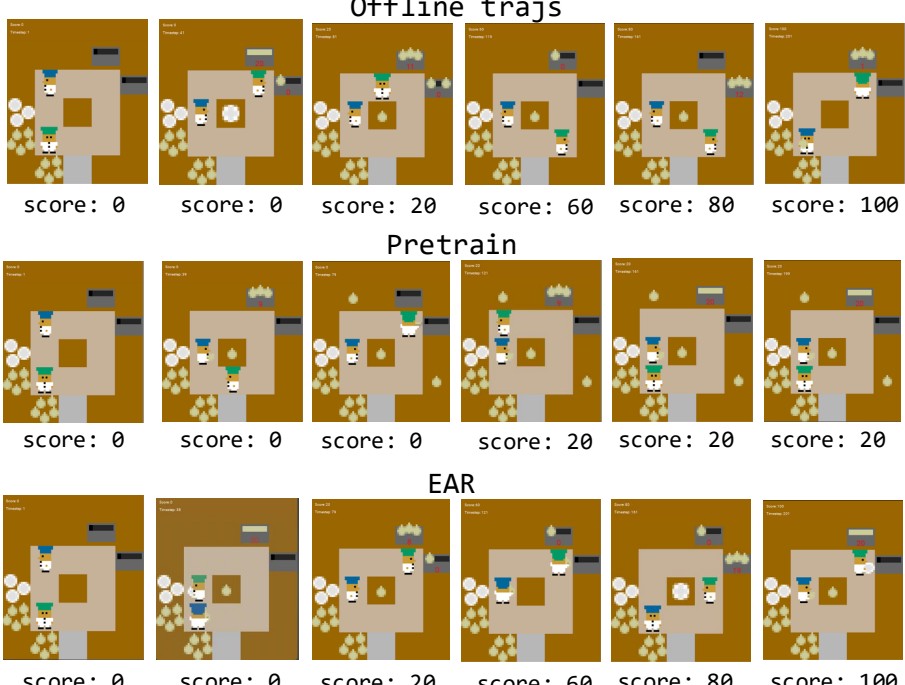

Figure 6: . Representative trajectory segments from three agents (offline dataset, pretrained agent, and EAR-trained agent) under style A in the Coordination Ring layout

Table 13: Performance across different preference styles. Each method is evaluated using Preference Score (↑) and Environment Return (↑). Results are reported on the layout *Coordination Ring*.

| Method | Style A | | Style B | | Style C | |
|---|---|---|---|---|---|---|
| | Pref. Score | Env. Return | Pref. Score | Env. Return | Pref. Score | Env. Return |
| Offline Trajs | 670.0 | 225.0 | 235 | 121 | 552 | 224 |
| MEP | $118.1 \pm 16.6$ | $35.6 \pm 6.2$ | $114.3 \pm 3.4$ | $49.9 \pm 3.7$ | $224.0 \pm 12.0$ | $85.5 \pm 5.3$ |
| TrajDi | $107.2 \pm 13.3$ | $31.1 \pm 5.5$ | $103.5 \pm 13.5$ | $39.4 \pm 8.1$ | $202.3 \pm 17.2$ | $75.5 \pm 8.3$ |
| **EAR** | $\mathbf{274.7} \pm 74.0$ | $\mathbf{88.6} \pm 31.0$ | $\mathbf{200.3} \pm 17.0$ | $\mathbf{103.6} \pm 9.4$ | $\mathbf{259.0} \pm 9.9$ | $\mathbf{110.3} \pm 3.3$ |

## F EAR Compared with Pretraining Methods Focused on Enhancing Diversity

Our approach specifically targets scenarios where a pretrained agent's generalization fails to adapt to a user's unique preferences—a situation commonly observed in both real-world and experimental settings, especially as environments become more complex (e.g., League of Legends, Honor of Kings, and Brawl Stars).

We acknowledge that when a biased human agent is unavailable, partner diversity becomes an important consideration and has been explored in several ZSC studies. However, the key insight of these diversity-oriented methods lies in maximizing environmental rewards through coordination with a diverse yet representative set of partners, rather than aligning with user-preference-related rewards. Consequently, when the biased human proxy's preferences fall outside the distribution covered by the pretrained policy pool, simply enlarging the diversity of policies during pretraining does not effectively resolve the misalignment issue.

To empirically validate this claim, we conducted experiments using two representative ZSC methods (MEP [Zhao et al., 2023] and TrajDi [Lupu et al., 2021]) that explicitly emphasize policy pool diversity. We selected pretrained checkpoints from ZSC-eval [Wang et al., 2024a] (built upon the HSP [Yu et al., 2023] framework), using three random seeds and a policy pool size of 36—larger than the pool used in our main experiments. When evaluated against the biased-preference human proxies in our benchmark. As shown in Table 13 these methods exhibited significantly weaker performance compared with EAR.

This finding supports our hypothesis that, when user preferences extend beyond the coverage of the pretrained policy pool, developing methods that leverage a small number of liked trajectories is both necessary and effective.

## G   Limitations

While our work demonstrates strong empirical performance, it has some natural limitations. Our method is relatively simple and could be further enhanced with more sophisticated designs. Additionally, we rely on biased preference proxies to simulate human preferences, which inevitably differ from real human behavior. We also do not fully exploit the abundant unlabeled trajectories left by human players, which could provide additional suboptimal data to further improve preference alignment. Despite these limitations, our approach establishes a solid foundation for learning cooperative agents that generalize to diverse partners while aligning with human preferences, and it opens promising directions for future exploration.

## H   Compute resources

All experiments were conducted on a single NVIDIA GeForce RTX 2080 Ti GPU. Table 14 reports the per-run execution time with the number of rollouts set to 200. We focus on the results for the Many Orders layout under Style A .

Table 14: Compute resources

| Methods | Execution time (s) |
| --- | --- |
| 1-epoch Behavior Cloning (BC) | 27.1 |
| One round of environment rollouts | 5.7 |
| PPO (15 epochs) after one rollout | 10.5 |

