# OpenReview forum: "Learning Preferences without Interaction for Cooperative AI: A Hybrid Offline-Online Approach"
_NeurIPS.cc/2025/Conference — NeurIPS 2025 poster_

### Official Review · Reviewer_5sd7 · 2025-06-29

**Clarity:** 2
**Significance:** 2
**Originality:** 3
**Rating:** 4
**Confidence:** 4

**Summary:**

This work proposes a method for integrating offline human data into RL agent training. They first pretrain using RL on a pool of existing policies before iterating a loop of single-epoch BC and online RL recovery (with existing policies). Their ablations demonstrate the importance of their method on balancing alignment with environment rewards.

**Questions:**

- In section 3.2, it seems like only "liked" trajectories are favored, but there is still critical information one can find from "unliked" trajectories. Why weren't negative trajectories included in your dataset and method?
- Why was BC chosen as the key method of alignment instead of inverse RL?
- Why are the bounds in algorithm 1 based on the original average returns and accuracy? It feels a bit arbitrary relative to a more formal function that takes average returns and alignment into account. For instance, I would consider KL-constrained RL to be a more formal way of combining RL with existing demonstrations.
- In lines 281-282, you say that a sub-optimal checkpoint is chosen to control for implicit preference alignment. How exactly is this checkpoint chosen then?

**Ethical Concerns:**

["NO or VERY MINOR ethics concerns only"]

**Final Justification:**

This work seems to stand in the middle ground between two popular lines of MARL work. On one side, we have zero-shot/ad-hoc coordination, where we assume no access to human data but instead must generalize to new partners. On another side, we have a large number of human demonstrations, which we can start from to do behavior cloning or do KL-constrained RL. This work instead imagines a setting where we have a limited amount of convention data, and instead wants to work with that specific convention at test time, which is more akin to few-shot coordination.

In this framing, the authors demonstrate the importance of few-shot coordination in the rebuttal by comparing against existing popular ZSC works, which their method outperforms. They do not explicitly compare against works that typically require large amounts of human demonstrations, which I feel would've neatly tied up loose ends for this work (though I would expect those methods to fall behind their approach). Overall, this work focuses on an understudied direction of MARL and provides a simple, yet effective method to use prior human data.

**Limitations:**

Yes (appendix E)

**Quality:**

2

**Strengths And Weaknesses:**

Strengths:
- The approach proposed in this paper is quite general and easy to implement.
- To my understanding, the proposed pipeline is novel.
- The experimental results support the claim that their technique balances environment reward with co-agent preferences.

Weaknesses:
- This work does not mention the recent body of work on generating diverse policies/conventions using cross-play minimization (LIPO/ADVERSITY/BRDiv/CoMeDi/L-BRDiv + others), which have been very successful at creating high-quality and diverse populations with no human data while still having strong ad-hoc teamwork with real humans because they try to be as robust as possible against the broadest set of policies. I expect comparisons against a representative subset of these methods in your experiments.
  - Additionally, it does not compare against existing RLHF work, like KL-constrained PPO (Way Off-Policy Batch Deep Reinforcement Learning of Implicit Human Preferences in Dialog), which operates on a very similar problem setting. It also does not compare against $PPO_{BC}$ from the original Overcooked work, which is also applicable to this problem setting.
- This work only conducts evaluations against human proxies, which are trained with RL from biased reward functions, instead of conducting a real human user study. This makes it hard to determine how well the proposed approach would work with real humans.


Other notes:
- The definition of ZSC used in this work is more aligned to the concept of ad-hoc teamwork (AHT) than ZSC. In particular, the original ZSC work (Hu et al., 2020) defines the goal of ZSC as finding a learning rule such that independent instances have high cross-play instead of defining based on an unknown set of human policies (which falls more in line with AHT).
- I think some of the equations have typos. In particular, I think there's an equal sign missing in equation 3 (before the second max), and there is a second max missing in equation 1.
- The text in figure 2 and 3 is too small relative to the text of the body, making it hard to read.
- In section 4.1, it is unclear what is meant by "pretraining" (Best reponse RL against the policy pool? BC on the policy pool? Some combination of both?) and where the policy pool comes from to begin with. I think it is important to explicitly say RL (and the specific algorithm) when training is mentioned, since pretraining in the context of LLMs typically refers to self-supervised learning.
- In table 3, I think all the results need to be multiplied by 100 (because you mean 73%, not 0.73%)

---

> ### Author Rebuttal · Authors · 2025-07-30
>
> Dear reviewer 5sd7:
>
> We sincerely thank you for the time and effort dedicated to reviewing our work. Your comments are highly appreciated and help us further improve the quality and clarity of our paper.
>
> We address each of your points in detail below:
>
> **Response to W1 (“work on generating diverse policies/conventions”):**
> We thank the reviewer for raising this important point. We will include a discussion of relevant methods on generating diverse policies and conventions in the related work section of the camera-ready version. Regarding the recent body of work on generating diversity through cross-play minimization, we selected two representative and validated methods from ZSC-eval [1]: TrajeDi [2] and MEP [3]. These methods have been shown to be effective in the Overcooked environment. For evaluation, we directly used the released checkpoints from ZSC-eval. Notably, these checkpoints were trained with a policy pool size of 36 (agents with standard preferences), which is larger than ours (18) and thus has stronger generalization potential. The experimental results on layout random1 are as follows:
>
> | Random1 | Style A (Pref. Score, Env. Return)| Style B (Pref. Score, Env. Return) |  Style C (Pref. Score, Env. Return) |
> |-----------|----------|----------|----------|
> | EAR      | **(274.7 ,  88.6)**    |  **(200.3 , 103.6)**  | **(259.0, 110.3)**
> | TrajDi | (100.0 , 32.5)     | (115.0 , 48.8)     | (190.0, 73.2)
> | Mep | (105 .0, 29.8)     | (112.0 , 45.3)     | (224.0, 82.4)
>
> These comparisons further support the necessity of explicit preference learning, as our method consistently performs well even when evaluated against strong ZSC baselines.
>
> **Further Response to W1 (on KL-constrained PPO and PPObc)**:
> In our setting, the number of available offline trajectories is extremely limited (around a dozen), making it difficult to estimate KL-constrained rewards reliably. Many states encountered during training are out-of-distribution (i.e., not present in the offline data), especially at the early stage. This leads to large estimation errors in KL-constrained methods due to the mismatch in dynamics between the offline trajectories and the policy pool policies. Our method avoids this issue by decoupling preference learning and generalization: the online recovery stage in EAR focuses solely on regaining generalization ability, without relying on preference alignment.
>
> Regarding PPObc, the core idea is to first fit a human proxy, and then train the cooperative agent via reinforcement learning to coordinate with this proxy. While this may work in simple or two-player settings, modeling the proxy accurately becomes significantly more challenging in multi-agent or more complex environments. In contrast, EAR does not rely on accurate modeling of the human proxy, and is able to simultaneously generalize and align with human preferences even when the dynamics in offline data and online recovery differ.
>
> **Response to W2 (“This work only conducts evaluations against human proxies”)**:
> We acknowledge this limitation. Due to budget and time constraints, we have not yet conducted experiments involving real human participants. We appreciate the reviewer’s feedback, and we consider this an important direction for future work.
>
>
> **Response to Notation Error (1)**:
> Thank you for pointing out the distinction between AHT and ZSC. We will include a discussion of their relationship in the camera-ready version. Our current use of ZSC terminology follows the convention adopted in prior works such as FCP, MEP, HSP, and ZSC-eval. That said, we agree that the definition may require further clarification, and we will revisit it carefully in our revision to ensure conceptual precision.
>
>
> **Response to Notation Errors (2), (3), and (5):**
> We appreciate the reviewer for identifying these issues. These are either minor typos or instances of unclear writing, and we will conduct a thorough review of the entire paper to eliminate such avoidable errors and improve the overall clarity.
>
>
> **Response to Notation Error (4):**
> Thank you for pointing out the imprecise use of the term “pretraining” in our paper. In our setting, pretraining refers to the process where the cooperative agent is trained via RL with policies from the policy pool to improve its generalization ability, i.e., its capacity to collaborate with a variety of agents to complete tasks in the environment. We acknowledge that this use of pretraining may differ from its usage in NLP, where pretraining typically involves learning from large-scale corpora to capture syntax and semantic knowledge. However, we view the intent as analogous—just as NLP models gain basic language competence through pretraining, our cooperative agent is meant to acquire foundational understanding of task structure and partner dynamics in Overcooked via this pretraining process. We will make this distinction clearer in the camera-ready version to avoid potential confusion.
>
> **Response to Q1:**
> Thank you for the question. We clarify that we did not use any poor-quality or noisy data in our experiments. However, we acknowledge that our current method does not leverage unlabelled or low-quality preference data, which we regard as a limitation and explicitly discuss in Appendix E. That said, a key contribution of our work is that EAR achieves a strong balance between generalization and preference alignment using only a small number of positive-labeled trajectories. We view this label-efficient approach as a meaningful step toward more practical preference learning in multi-agent settings.
>
>
> **Response to Q2 and Q3:**
> In our current setup, we do not attempt to model a preference-based reward function. The primary reason is the extremely limited amount of offline preference data—only around 10-20 trajectories. Under such sparse data, reward learning methods (e.g., inverse reinforcement learning) are prone to extrapolation errors due to a large number of out-of-distribution states, which leads to unreliable or unstable reward estimation.
>
> Similarly, using KL-constrained RL to learn from these offline trajectories faces challenges: the dynamics of the offline trajectories differ significantly from those induced by the policy pool, resulting in large initial KL divergence and unreliable gradients, especially early in training. This can cause performance degradation due to compounding errors. In contrast, our method avoids this issue by separating preference alignment and generalization: EAR’s online recovery phase is used solely to restore generalization ability, while preference information is distilled only from the limited set of labeled trajectories.
>
>
>
> **Response to Q4:**
> In the main paper, “checkpoint” refers to the best-performing checkpoint of FCP. The term sub-optimal is used in two senses:
> (1) Under the FCP setup, the biased human agent used for supervision is no longer available, and its behavior closely resembles the offline preference trajectories. This suggests that there is limited room for improvement beyond the offline data.
> (2) More broadly, in the absence of the human supervisor, other methods from the ZSC literature such as TrajeDi or MEP may achieve better coordination than FCP when evaluated under standard ZSC metrics. However, when evaluated under our preference alignment objective, these methods do not significantly outperform FCP, suggesting that coordination alone is insufficient for preference learning.
>
> If you have any additional questions or concerns, we would be happy to provide further clarification and are glad to have the opportunity to elaborate more on our work.
>
>
> [1] ZSC-Eval: An Evaluation Toolkit and Benchmark for Multi-agent Zero-shot Coordination (NeurIPS 2024)
>
> [2] Trajectory Diversity for Zero-Shot Coordination. (PMLR 2021)
>
> [3] Maximum Entropy Population-Based Training for Zero-Shot Human-AI Coordination (NeurIPS Cooperative AI Workshop 2021)

---

> ### Comment · Reviewer_5sd7 · 2025-08-03
>
> Thank you for the clarifications and detailed rebuttal. I still feel that other comparisons would strengthen the paper, but the TrajeDi and MEP results are promising (though to clarify these are not "cross-play minimization" techniques, but instead achieve diversity in different ways).
>
> I think the paper would've been stronger if it compared with the existing RLHF work (KL-constrained PPO and PPO-bc), especially comparing the performance versus number of human samples, but I think this work is acceptable as it stands so I will raise my score.

---

> > ### Author Response · Authors · 2025-08-04
> >
> > Dear reviewer 5sd7:
> >
> > Thank you for taking the time and effort to review our work. We truly appreciate your feedback and will carefully incorporate your comments to improve the paper in the camera-ready version.
> >
> > Sincerely,
> >
> > Authors

---

### Official Review · Reviewer_7F41 · 2025-07-02

**Clarity:** 3
**Significance:** 2
**Originality:** 2
**Rating:** 4
**Confidence:** 4

**Summary:**

This work proposes a hybrid RL training framework that alternates between offline preference learning and online generalization recovery for cooperative agents. The cooperative AI problem setup assumes pretrained policies for cooperative agent training, and availability of offline preference data. The paper attempts to simultaneously enhance the zero-shot generalization and preference alignment capabilities of the cooperative agent. The training regime and metrics are validated in the Overcooked environment against a number of relevant baselines.

**Questions:**

1) What are the conditions on the set of pretrained policies available to the agent? How much should policies be reflective of/adhere to the preferences in the offline data?
2) What exactly does lightweight behavior cloning mean in the training regimen?

**Ethical Concerns:**

["NO or VERY MINOR ethics concerns only"]

**Final Justification:**

Most of my concerns and questions have been addressed by the rebuttal clarifications. I believe the work presents an interesting tweak with useful empirical results. However, I still have reservations about the impact of this work. So, I decided to maintain my borderline accept rating, but with improved confidence.

**Limitations:**

The limitations of the training paradigm and dataset collection need to be stated.

**Paper Formatting Concerns:**

1) Figure 2: Y-axis scales need to be changed for each style. Currently, it does not seem like all styles should be on the same Y-axis scale.

**Quality:**

3

**Strengths And Weaknesses:**

### Strengths
1) The paper is largely clear and well-structured.
2) The work proposes an interesting new tweak of epoch-wise alternation recovery, and provides empirical experimental evidence of its success.

### Weaknesses
1) The post-match "like" assumption is informative at the trajectory level, but may not be so at the granular action level. The trajectory is endorsed by the human, but the human may still have corrections or different preferences for some individual actions at specific states. This may affect zero-shot coordination.

2) The training method of epoch-wise alternation recovery proposed involves performing generalization recovery with the policy pool until either the average return is greater than the original value or the budget is exhausted. This can be computationally expensive, considering recovery is generally slow. It would be useful to get information on the budget limit used, the number of times the budget was exhausted, and the average time taken for recovery, etc. In addition, how would these change for a more diverse policy pool, etc.

3) Section 5.3 needs to be explained with respect to preference styles. Currently, it's too generic and not well-grounded in the experimental results.

---

> ### Author Rebuttal · Authors · 2025-07-30
>
> Dear reviewer 7F41:
>
> We sincerely thank you for the time and effort dedicated to reviewing our work. Your comments are highly appreciated and help us further improve the quality and clarity of our paper.
>
> We address each of your points in detail below:
>
>
>
> **Response to W1 (“human may still have corrections or different preferences”):**
> We agree with the reviewer’s observation. Indeed, the fact that “a trajectory is endorsed by the human” does not imply that the human will always make identical decisions in the future. We have taken this uncertainty into account when designing our benchmark. Specifically:
>
> 1.	To encourage stochasticity and avoid overfitting to specific behaviors, we generate and evaluate offline trajectories by sampling from parameterized agents, rather than using deterministic policies.
>
> 2.	As stated in Lines 267–269 of the paper: “we retain the trajectories whose biased return reaches at least 80% of the maximum return under that preference, and treat them as liked trajectories. The corresponding human-side policies in these trajectories serve as test-time collaborators for evaluating the cooperative agent.” Each human preference is associated with a well-defined reward function, consistent with the assumption made in ZSC-eval [1]. Multiple human proxy agents may be used to evaluate under the same preference, further capturing the diversity and variability of human decision-making in real-world settings.
>
> **Response to W2 (Concern about computational cost):**
> We appreciate the reviewer’s concern regarding computational efficiency. In our experiments, we set the rollout size to 200 and the trajectory length to 400, with a total environment step budget of 20 million. This implies that EAR performs at most 250 rollouts throughout training.
>
> All experiments were conducted on a single NVIDIA RTX 2080 Ti GPU. Under the random1 layout, style A, the average training time per step is as follows:
>
> | Random1 (methods in EAR)| Execution time (s) |
> |-----------|----------|
> | 1-epoch Behavior Cloning (BC)    | 22.3
> | One round of environment rollouts (rollout size 200)| 27.2
> | PPO (15 epochs) after one rollout | 9.4
>
> | Random1| The number times rollouts was exhausted (max 250)|
> |-----------|----------|
> |Style A|204.3 |
> |Style B|188.3|
> |Style C|205.6|
>
> As shown in the table above, EAR requires approximately 2 to 4 hours of execution time per style on a single RTX 2080 Ti GPU.
>
> Regarding the question “how would these change for a more diverse policy pool”, we sincerely thank the reviewer for raising this important point. Exploring the scalability of our method under a more diverse policy pool is a valuable direction, and we consider it an important aspect for future work.
>
> **Response to W3 (Unclear description in Section 5.3):**
> We apologize for the unclear description in Section 5.3. Our intent was to first present the overall performance of our method, and then to provide a deeper understanding of the learned preferences by visualizing event-count vectors (one per trajectory) using UMAP. The resulting clusters reflect the impact of preference learning on the cooperative agent’s behavior.
> We appreciate the reviewer’s feedback, and we will improve this section by including more fine-grained, preference-specific analyses in the camera-ready version.
>
> **Response to Q1:**
> The pretrained policies available to the agent—i.e., the policies in the policy pool—are not subject to any explicit conditioning or constraints. In our experimental setup, these policies are trained via self-play under the standard environment reward function, meaning their behavior is solely driven by reward maximization. This design choice ensures that no information about human preferences is leaked into the policy pool, maintaining a clear separation between environment reward and human preference.
> Regarding “how much should policies be reflective of / adhere to the preferences in the offline data,” we determine whether a policy aligns with a given preference by computing the preference score of the trajectories it generates. This allows us to assess how well a given policy adheres to a particular reward function without requiring explicit conditioning.
>
> **Response to Q2:**
> By lightweight behavior cloning, we refer to the fact that in our method, the behavior cloning (BC) phase requires significantly less time and computational resources compared to the online recovery stage.
>
> If you have any additional questions or concerns, we would be happy to provide further clarification and are glad to have the opportunity to elaborate more on our work.
>
> **Response to Paper Formatting Concerns:**
> Thank you for pointing out the formatting concerns. We will carefully review all figures and thoroughly check the entire paper to ensure that all formatting issues are resolved in the camera-ready version.
>
> If you have any additional questions or concerns, we would be happy to provide further clarification and are glad to have the opportunity to elaborate more on our work.
>
>
> [1] ZSC-Eval: An Evaluation Toolkit and Benchmark for Multi-agent Zero-shot Coordination (NeurIPS 2024)

---

> > ### Comment · Reviewer_7F41 · 2025-08-08
> >
> > I thank the authors for their explanations, comments, and clarifications. Most of my concerns and questions have been addressed.
> > As stated in my review, I believe the work presents an interesting tweak with useful empirical results. However, I still have reservations about the impact of this work. So, I have decided to maintain my borderline accept rating, but with improved confidence.

---

> > > ### Author Response · Authors · 2025-08-08
> > >
> > > Dear reviewer 7F41:
> > >
> > > Thank you again for taking the time and effort to review our work. We truly appreciate your feedback and will carefully incorporate your comments to improve the paper in the camera-ready version.
> > >
> > > Sincerely,
> > >
> > > Authors

---

### Official Review · Reviewer_7shS · 2025-07-07

**Clarity:** 3
**Significance:** 3
**Originality:** 2
**Rating:** 5
**Confidence:** 4

**Summary:**

At a high level, this work addresses the problem of incorporating human preference data into the training of policies for AI agents that can cooperate with previously unseen partners.  They first pre-train a cooperative policy against a pool of human proxy policies learned using the previously described Fictitious Co-Play framework.  They then allow (simulated) humans to interact with this policy, and "like" the joint trajectories that they found to be most successful.  Finally, they alternate between training (under a behavioral cloning loss) on these "liked" trajectories, and training against the human proxy pool.  The purpose of this alternating training scheme is to avoid catastrophic overfitting to the small set of human-annotated trajectories.  Experimental results on a subset of the Overcooked zero-shot coordination benchmark show that this method significantly outperforms the FCP baseline against "stylistically biased" human partners.  Ablation experiments also show that removing the alternating training setup significantly degrades this performance.

**Questions:**

1. For the HSP-based policies used to generate the trajectories for behavioral cloning, to what extent were the learned policies "sub-optimal" with respect to the base Overcooked reward function?  Ideally, methods like FCP would cover the space of near-optimal joint strategies, so perhaps relatively large reward pertubations could explain its poor performance with the "biased" partners?
2. How well did the joint strategies trained with FCP, and the pre-trained policy trained against them, perform under the base Overcooked reward (without perturbations)?
3. Was the architecture of the pre-trained policy recurrent (allowing for adaptation)?

**Ethical Concerns:**

["NO or VERY MINOR ethics concerns only"]

**Final Justification:**

The additional experiments the authors discuss, and their further clarifications about the method, have addressed most of my concerns, and I have increased my score accordingly.

**Limitations:**

Yes

**Paper Formatting Concerns:**

N / A

**Quality:**

3

**Strengths And Weaknesses:**

**Strengths:**
1. The proposed EAR framework proved to substantially outperform the FCP baseline, using only a small set of style-specific trajectories.
2. Possibly the most interesting result from this work is in Table 1, showing that alternating between BC and "recovery" training against the FCP pool leads to significantly improved performance over a single round of BC + recovery.

**Weaknesses:**
1. HSP is used to generate stylistically biased human trajectories, but is not used as a baseline method itself.  It would seem likely that a policy pre-trained against a pool of agents generated through HSP (trained on perturbed reward functions) would outperform FCP in this setting, even without BC finetuning or alternating recovery.
2. There are a number of other ZSC methods that might been good baselines here, for example TrajeDi (Lupu et al. 2021) or LIPO (Charakorn et al., 2023).  The poor performance of FCP pre-training in these experiments likely has to do with the diversity of the human partners the method is evaluated against.  Compared to FCP, both TrajeDi and LIPO (along with several other baselines) have mechanisms to encourage diversity within the pool of target agents, and might be expected to perform well even without fine-tuning on human preferences.

---

> ### Author Rebuttal · Authors · 2025-07-30
>
> Dear reviewer 7shS:
>
> We sincerely thank you for the time and effort dedicated to reviewing our work. Your comments are highly appreciated and help us further improve the quality and clarity of our paper.
>
> We address each of your points in detail below:
>
> **W1: HSP as a Baseline**
>
> Our work aims to simulate a realistic setting where we are given an available policy pool and environment, along with only a very limited number of biased human trajectories (possibly as few as a dozen). The key goal of our method, EAR, is to effectively infer the preferences of a biased human agent—even when this agent is not part of the policy pool—using only a small amount of offline data. This setup closely mirrors practical constraints in real-world game industry scenarios. In contrast, in the Overcooked environment, if the biased human agent is also available (i.e., present in the policy pool), then the online recovery phase would encounter the same dynamics as the offline trajectories. This setting is already well addressed by existing work such as ALIGN-GAP [1]. Our work instead tackles a more challenging and realistic setting that has not been adequately addressed by prior methods.
>
> | Random1 | Style A (Pref. Score, Env. Return)| Style B (Pref. Score, Env. Return) |  Style C (Pref. Score, Env. Return) |
> |-----------|----------|----------|----------|
> | EAR      | **(274.7 ,  88.6)**    |  **(200.3 , 103.6)**  | **(259.0, 110.3)**
> | ALIGN-GAP | (145.0 , 50.0)     | (131.0 , 58.8)     | (235.0, 100.0)
>
> **W2: ZSC Methods:**
>
> We agree that in cases where the biased human agent is unavailable, partner diversity becomes important and has been considered in some ZSC works. However, it is important to note that these methods typically aim to maximize environment reward, not to align with human preferences—which is the central goal of our preference learning setup.
>
> To provide a fair comparison, we reproduced TrajeDi [3] and MEP [2], two representative ZSC methods. For evaluation, we directly used publicly released checkpoints from ZSC-eval [4]. These checkpoints were trained with a policy pool of size 36 (agents with standard preferences), which is larger than ours (18), and thus exhibits stronger generalization ability. The experimental results are summarized below:
>
> | Random1 | Style A (Pref. Score, Env. Return)| Style B (Pref. Score, Env. Return) |  Style C (Pref. Score, Env. Return) |
> |-----------|----------|----------|----------|
> | EAR      | **(274.7 ,  88.6)**    |  **(200.3 , 103.6)**  | **(259.0, 110.3)**
> | TrajDi | (100.0 , 32.5)     | (115.0 , 48.8)     | (190.0, 73.2)
> | Mep | (105 .0, 29.8)     | (112.0 , 45.3)     | (224.0, 82.4)
>
> These results further demonstrate the necessity of explicitly modeling preference learning, beyond general reward maximization.
>
> **Response to Q1**:
> In the main text, the term “sub-optimal” is used in two respects. First, in the context of FCP, the biased human agent is unavailable, and its performance in the ORACLE setting is close to that of the offline trajectories—indicating limited improvement. Second, we acknowledge that in the absence of the biased human, methods such as TrajeDi and MEP from the ZSC literature may perform better than FCP in terms of coordination. However, their performance does not show significant advantages over FCP when evaluated under the preference learning objective.
>
> The relatively poor performance of FCP in our benchmark is indeed due to its inability to cover the space of the biased human, which is a well-known challenge in real-world game industry applications. Human preferences are inherently diverse and difficult to fully anticipate or enumerate. Regarding the “large reward perturbations”: the implicit reward induced by human preferences can deviate substantially from the environment reward. More fundamentally, since the biased human is unavailable, the dynamics encountered during online recovery differ significantly from those observed in the offline trajectories. This distribution shift makes it difficult for agents trained via FCP to adapt effectively to the biased human’s behavior.
>
> **Response to Q2**:
> Below, we provide the average environment return of the agent trained using FCP across its policy pool, as well as its scores under the three different human preference styles:
>
> | Random1 | Style A (Pref. Score)| Style B (Pref. Score) |  Style C (Pref. Score) |Env. Return|
> |-----------|----------|----------|----------|----------|
> |FCP |209.0|-55.0|244.0|142.0|
>
> **Response to Q3**:
> In principle, the pre-trained policy can be recurrent. However, in our experiments, we opted for a non-recurrent architecture due to implementation considerations when applying Behavior Cloning. Training with recurrent policies requires handling memory, which involves either storing hidden states for each timestep or adopting a trajectory-wise training scheme. We appreciate your thoughtful question, and we will consider extending our experiments to include recurrent architectures in future work.
>
> If you have any additional questions or concerns, we would be happy to provide further clarification and are glad to have the opportunity to elaborate more on our work.
>
> [1] ONLINE-TO-OFFLINE RL FOR AGENT ALIGNMENT. (ICLR 2025)
>
> [2] Maximum Entropy Population-Based Training for Zero-Shot Human-AI Coordination (NeurIPS Cooperative AI Workshop 2021)
>
> [3] Trajectory Diversity for Zero-Shot Coordination. (PMLR 2021)
>
> [4] ZSC-Eval: An Evaluation Toolkit and Benchmark for Multi-agent Zero-shot Coordination (NeurIPS 2024)

---

> > ### Comment · Reviewer_7shS · 2025-08-05
> > **Response to Rebuttal**
> >
> > Thank you for running the additional experiments, that does clarify things a bit.
> >
> > As I understand the problem setting, the base policy is trained without access to the "true" reward function that the human-AI team will be evaluated under.  The question here is whether the AI policy can adapt to this unknown reward function (a perturbation of the original reward) using only a small amount of human preference data.  On this unknown reward, EAR significantly outperforms FCP and the other baselines.
> >
> > I do suspect however that in the Overcooked task you might see good performance under even the perturbed reward if you used recurrent policies that were allowed to adapt to their partners, as the perturbations primarily serve to generate different "styles" of solutions.

---

> > > ### Author Response · Authors · 2025-08-06
> > >
> > > Dear reviewer 7shS:
> > >
> > > Thank you again for taking the time and effort to review our work.
> > >
> > > In response to your question regarding the use of recurrent policies that are allowed to adapt to their partners: While EAR itself does not use recurrent policies, the checkpoints of two ZSC methods we selected for evaluation in ZSC-eval—MEP and TrajDi—do utilize recurrent policies and are specifically designed to handle the diversity of human partners. Our results show that, in the absence of any information about perturbed rewards, even these recurrent policies struggle to achieve good performance when relying solely on diversity.
> > >
> > > Nevertheless, we appreciate your insightful suggestion and will consider incorporating a more in-depth discussion on this topic. We will further revise our paper based on your comments in the camera-ready version.
> > >
> > > Sincerely,
> > >
> > > Authors

---

> > > > ### Comment · Reviewer_7shS · 2025-08-06
> > > > **Response**
> > > >
> > > > Ah, thank you for clarifying, that makes more sense, and I will update my review accordingly.

---

> > > > > ### Author Response · Authors · 2025-08-07
> > > > >
> > > > > Dear reviewer 7shS:
> > > > >
> > > > > We sincerely appreciate your time, constructive comments, and willingness to engage in this discussion. Your feedback has been very helpful, and we will carefully reflect it in the camera-ready version.
> > > > >
> > > > > Sincerely,
> > > > >
> > > > > Authors

---

### Official Review · Reviewer_YpWX · 2025-07-14

**Clarity:** 2
**Significance:** 3
**Originality:** 1
**Rating:** 4
**Confidence:** 4

**Summary:**

The paper addresses the conflict between generalizing to unseen partners and aligning with limited offline preference data in cooperative reinforcement learning. It introduces a training strategy named Epoch-wise Alternation Recovery, which alternates between one epoch of lightweight behavior cloning and generalization recovery. This strategy enables cooperative agents to maintain strong generalization capabilities while effectively incorporating human preferences. Experiments in the Overcooked environment show that the new strategy achieves both high preference scores and high environment returns, demonstrating that it effectively balances the two objectives.

**Questions:**

The experimental comparison would be more convincing if it included at least one method from section 2.3 that jointly addresses generalisation and preference alignment.
The exposition in Section 4 breaks the announced flow: subsection 4.2 inserts design rationale before stage 2 is introduced. Re-ordering the material so that Stages 1–3 are described sequentially, followed by the design discussion, would improve readability and align with the section’s opening statement.
Several table annotations need clarification. Please specify what the “Offline Trajs” row in Table 1 represents, and ensure the description of Table 2 is consistent between the caption and the main text. Resolving these discrepancies would remove residual ambiguity in the presentation.

**Ethical Concerns:**

["NO or VERY MINOR ethics concerns only"]

**Limitations:**

yes

**Paper Formatting Concerns:**

see above

**Quality:**

4

**Strengths And Weaknesses:**

Pros.

Tackling distribution shift in human‑cooperative reinforcement learning is a timely and underexplored problem. This fresh solution is poised to spark follow‑up studies on generalization and preference alignment under non‑stationary conditions.
I appreciate that the paper’s core claims rest on a solid foundation: the methods are both principled and well matched to the problem, and the extensive suite of experiments convincingly demonstrates their effectiveness. 3. The three‑stage training pipeline is logically coherent and neatly justified. The paper articulates the rationale for this integration clearly, and the differentiation from prior work is precise, with all key citations in place.
The paper is well written and supplies enough implementation detail to reproduce results.
Cons.

The promise to “describe each stage in detail below” is undermined when subsection 4.2 veers into design rationale instead of covering the second stage. This break in flow makes it harder to follow the pipeline in the order the authors intended.
There’s confusion around Table 2: the main text calls out “benchmark proxy agents not involved in the offline preference data,” yet the caption refers to “all benchmark human proxy agents.” Aligning these descriptions is crucial for readers to understand exactly what’s being evaluated.
Although the experiments are thorough, the baseline set skips over promising alternatives—such as methods leveraging successor features or contrastive learning for joint generalization and preference alignment. Including those would strengthen the comparative evaluation and better position the contribution relative to state‑of‑the‑art approaches.

---

> ### Author Rebuttal · Authors · 2025-07-30
>
> Dear reviewer YpWX,
>
> We sincerely thank you for the time and effort spent reviewing our paper and for providing valuable suggestions for improvement. We address each of the comments point by point below.
>
> **Clarification on Section 4.2**
> We apologize for the confusion caused by unclear descriptions in Section 4.2 and the corresponding tables. The sentence “describe each stage in detail below” was intended to introduce the methodological description of the three stages of our approach. Specifically, the second stage involves a single round of online generalization recovery after offline behavior cloning. In Section 4.2, our goal was not only to present the detailed implementation of this second stage, but also to analyze its limitations through experiments—thus motivating the necessity of the third stage: Epoch-wise Alternation Recovery. We acknowledge that the current title of Section 4.2 may have been misleading, and we commit to revising it in the camera-ready version to better reflect the section’s content.
>
> **Clarification on Table Descriptions**
> In Table 1, offline trajs refer to the limited number of offline expert trajectories available during training. These serve as golden trajectories and are generated by preference-aligned agent pairs that have reached a Nash equilibrium through self-play within the benchmark environment. We also apologize for the ambiguity in our phrasing of “agents that were not involved in the offline preference data” and “all benchmark human proxy agents.” Our intention was to convey that, in addition to evaluating on the agents that were involved in the offline preference data, we also conducted evaluations on agents that were not involved in the offline data, in order to assess the generalization ability of the cooperative agent. The combination of results from both sets of agents constitutes what we refer to as “all benchmark human proxy agents,” which is exactly what is reported in Table 2. We will unify and clarify the terminology in the table caption in the camera-ready version to avoid ambiguity.
>
> **On the Choice and Applicability of Baselines**
> The methods discussed in Section 2.3 are designed for different RL settings. In our proposed setup, the task is cooperative in nature, where each partner induces a different environment dynamic. In contrast, most baselines in Section 2.3 assume that the online environment shares the same dynamics as the offline trajectories—an assumption that does not hold in our case, and which limits their direct applicability. Nevertheless, we made efforts to reproduce ALIGN-GAP [1] (published at ICLR 2025) within our setting.  For reward model learning, we used the ground-truth reward function directly. Specifically, when the preference-based reward of a trajectory exceeds the threshold observed in the offline trajectories, the reward model outputs a “like” signal for that trajectory. The reproduced results under the **random1** layout are as follows:
>
> | Method | Style A (Pref. Score, Env. Return)| Style B (Pref. Score, Env. Return) |  Style C (Pref. Score, Env. Return) |
> |-----------|----------|----------|----------|
> | EAR      | **(274.7 ,  88.6)**    |  **(200.3 , 103.6)**  | **(259.0, 110.3)**
> | ALIGN-GAP | (145.0 , 50.0)     | (131.0 , 58.8)     | (235.0, 100.0)
>
> It is important to note that the ALIGN-GAP method was originally designed for single-agent settings, where the dynamics of the offline trajectories and the online environment are assumed to be the same. This key difference from our cooperative multi-agent setting, where the dynamics vary depending on the partner, makes the direct application of ALIGN-GAP ineffective. The experimental results further highlight the necessity of adopting our proposed approach.
>
> If you have any additional questions or concerns, we would be happy to provide further clarification and are glad to have the opportunity to elaborate more on our work.
>
> [1] ONLINE-TO-OFFLINE RL FOR AGENT ALIGNMENT.  (ICLR 2025)

---

### Decision · Program_Chairs · 2025-09-17

**Decision:**

Accept (poster)

**Comment:**

This paper proposes a new framework for cooperative AI that uses a small amount of preference data while adapting online.
The work is timely and novel, and experiments show that the method is simple yet effective. In particular, the evaluation in the overcooked environment demonstrates clear improvements in both preference scores and environment rewards, which was highlighted as a major strength.

There were concerns about unclear presentation, limited baselines, and the reliance on proxy agents rather than real humans.
The authors addressed these points with additional experiments and revisions, and showed that the proposed method outperforms existing methods despite its simplicity.

Overall, the paper combines novelty and empirical support, and points to an important direction for cooperative AI research.